# A case-control study evaluating CT signs of xiphoid process associated with xiphodynia

**Ryosuke Ono**[1,2]*, **Ken Horibata**[1,2]

**1** Department of Community Medicine, Kameyama, Mie University School of Medicine, Tsu, Mie, Japan,
**2** Department of Internal Medicine, Kameyama Municipal Medical Center, Kameyama, Mie, Japan

* rikusan2005@yahoo.co.jp

## Abstract

This study assessed whether CT signs of the xiphoid process, such as the xiphisternal angle and soft tissue compression, are useful for diagnosing xiphodynia. Conducted as a case-control study within a cohort, it involved 1560 participants who visited a small urban hospital in Japan for chest or abdominal pain between January 2021 and September 2023. From this group, patients who underwent CT scans that included the xiphoid process were selected. The study group consisted of nine individuals diagnosed with xiphodynia, while the control group included 321 individuals diagnosed with other causes of pain. No significant differences were found in the xiphisternal angle, soft tissue compression, or xiphoid tip features between the groups. However, in about 70% of cases, the xiphoid process curved forward and then backward. These findings suggest that the xiphisternal angle is not a useful marker for diagnosing xiphodynia, and that the curvature of the xiphoid process is common regardless of the condition.

## Introduction

Xiphodynia is a relatively rare condition characterized by pain originating from the xiphisternal joint or the xiphoid process. Although it does not typically progress to a severe state, prolonged pain can lead to a decrease in quality of life [1]. Symptoms of xiphodynia often manifest as chest pain or upper abdominal pain, but may also include radiating pain to the back, neck, or upper limbs. Due to these symptoms, it is commonly misdiagnosed as other conditions such as cardiovascular or gallbladder diseases. Additionally, due to low awareness of the condition, it frequently goes undiagnosed or treatment is delayed [1]. While there is no established treatment, local injection of 40mg depot methylprednisolone and 50mg lidocaine at the most tender area on the xiphoid process has been reported to be effective [2]. In cases where there is poor response to local injection or in refractory cases, xiphoid process excision surgery is considered as an option [1].

**Data availability statement:** The minimal anonymized dataset for this study can be found at https://datadryad.org/dataset/doi:10.5061/dryad.ghx3ffbxw or the DOI: 10.5061/dryad.ghx3ffbxw.

**Funding:** The author(s) received no specific funding for this work.

**Competing interests:** The authors have declared that no competing interests exist.

While diagnostic criteria for xiphodynia have not been established, it is diagnosed by the presence of reproducible tenderness upon palpation of the xiphoid process [1,3]. As part of diagnostic treatment, injecting local anesthetic agents directly onto the xiphoid process can help relieve pain [4]. Additionally, organic diseases such as ischemic heart disease, gallbladder disorders, and gastroesophageal reflux disease have been reported to present similar symptoms, underscoring the importance of excluding these organic diseases [3,5].

It has been suggested that anterior protrusion of the xiphoid process is a risk factor for xiphodynia, and several reports suggest that using imaging tests to look at the xiphoid process from the side can help diagnose xiphodynia when it sticks out forward [6–9]. The largest study conducted to date was reported by Maigne et al., who compared the xiphisternal angle between lateral CT images of the xiphoid process in three patients diagnosed with xiphodynia and 60 healthy individuals without xiphodynia. The results showed that the xiphisternal angles were 105°, 120°, and 135° in xiphodynia patients, while it was $172 \pm 14°$ in the healthy group. Therefore, they suggested that a decrease in the xiphisternal angle could be useful for diagnosing xiphodynia, proposing the possibility of inflammation around the xiphoid process due to anterior curvature being involved in the pathophysiology of xiphodynia [6]. However, due to the small sample size of three cases, the diagnostic accuracy was not clearly established, and even in the healthy group, 14 out of 60 individuals exhibited a decrease to 140–159°. Furthermore, in our clinical experience, we often encounter patients with significant decreases in the xiphisternal angle but without symptoms. Additionally, we have reported cases of xiphodynia with a xiphisternal angle of 155°, which does not show a pronounced decrease [10]. Therefore, we question the utility of the xiphisternal angle for diagnosing xiphodynia.

Additionally, we believe that signs of the xiphoid process compressing anterior soft tissue or anatomical features at the tip of the xiphoid process may be useful for diagnosing xiphodynia. The anterior surface of the xiphoid process is attached to the rectus abdominis muscle, with subcutaneous tissue and skin lying anteriorly [3]. Evidence of the xiphoid process tip compressing anterior soft tissue, including the rectus abdominis muscle, can be observed in many of the presented images in the literature on xiphodynia [6–8,10]. However, we could not find any reports investigating these details.

In this study, we aim to evaluate whether CT signs such as xiphisternal angle and evidence of soft tissue compression are useful for the imaging diagnosis of xiphodynia.

## Methods

### Study design

A case-control study within a cohort.

### Population (Fig 1)

This study targeted patients who visited the internal medicine outpatient department of Kameyama Municipal Medical Center between January 2021 and September 2023

with complaints of chest pain or abdominal pain. Case group were defined as patients diagnosed with xiphodynia who underwent CT scans including the xiphoid process. Diagnosis of xiphodynia was defined as meeting all of the following criteria: (1) presence of chest or upper abdominal pain, (2) reproducible tenderness upon palpation of the xiphoid process, (3) absence of other conditions more likely than xiphodynia as the cause of pain, and (4) improvement of pain with local injection of anesthetic agents onto the xiphoid process or xiphoid process excision surgery. Control groups were defined as patients who underwent CT scans including the xiphoid process and were diagnosed with other conditions than xiphodynia as the cause of pain. Patients with unknown causes of pain were excluded from the control group.

### sample size

It was calculated based on a case-control study by Maigne et al. regarding the xiphoid process sternal angle [6]. Using a factor of the xiphoid process sternal angle being less than 160°, the assumed proportion of cases with this factor was 0.8. Given that 14 out of 60 cases were less than 160° based on previous studies, the proportion for the control group was set at 0.23. With an α error of 0.05, power of 0.8, and a case-to-control ratio of 1:10, the calculation resulted in 7 cases and 70 controls.

### Data collection

Patient data were accessed for research purposes from 01/10/2023 to 31/03/2024. Medical records were reviewed to extract demographic information including age, gender, site of pain, and underlying cause of pain.

### Measurement of CT signs

CT examinations were utilized to measure signs of interest. Sagittal images were created, incorporating the xiphoid process from the thinnest axial section.

The xiphisternal angle was measured using a method that we defined due to its absence in previous literature. We identified two patterns for measurement: one between the xiphoid process and the sternal body angle, and the other

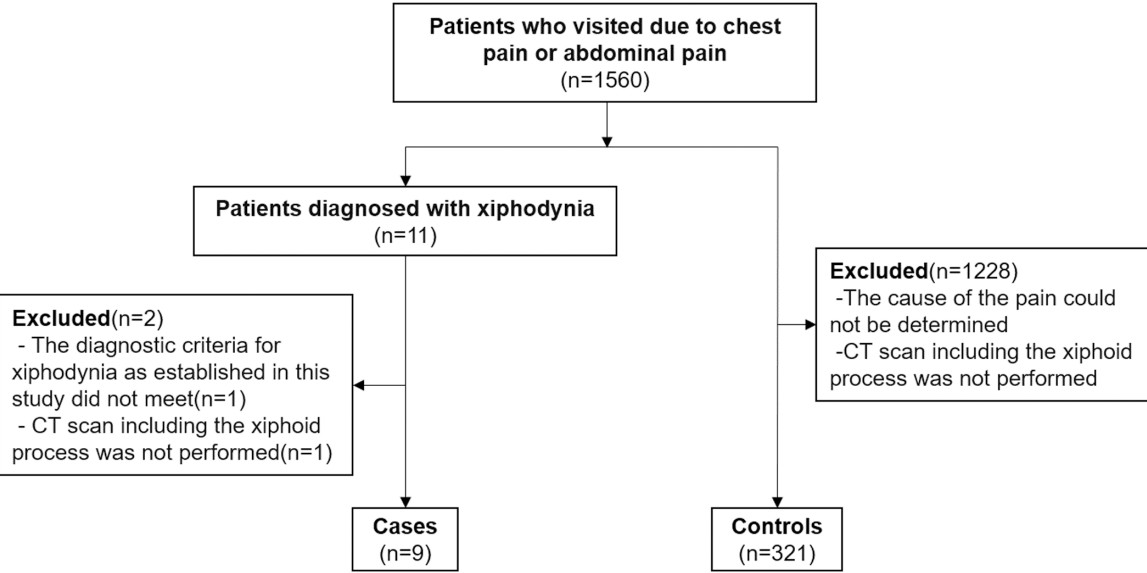

**Fig 1.  Case and control groups inclusion flowchart.**

between the xiphoid process and the base of the xiphoid process [6–9]. Additionally, variations in the xiphoid process, such as bending midway or inversion, were considered, leading to the definition and measurement of four angles: the xiphoid-sternal body angle (XSBA), xiphoid to xiphoid base angle (XXBA), xiphoid tip to sternal body angle (XTSBA), and tip to base angle of the xiphoid process (TBAXP), as illustrated in Fig 2.

Regarding compression signs caused by the xiphoid process, we assumed and measured the anterior shift of the rectus abdominis (ASRA), anterior shift of the skin surface (ASSS), and thinness of the subcutaneous tissue (TST) as useful indicators (Fig 3). As for anatomical signs at the tip of the xiphoid process, we measured the penetration of the xiphoid tip into the rectus abdominis (PXTRA), calcification of the xiphoid tip (CXT), and hypertrophy of the rectus abdominis (HRA) in contact with the tip of the xiphoid process (Fig 3). The definitions for each sign are as follows: (1) ASRA: At the most ventral projection of the xiphoid process, the rectus abdominis shifts anteriorly. (2) ASSS: At the most ventral projection of the xiphoid process, the skin surface line shifts anteriorly. (3) TST: Subcutaneous tissue compressed by the xiphoid process is thinner than surrounding tissue. (4) PXTRA: The tip of the xiphoid process contacts the rectus abdominis, and the tip and rectus abdominis are not parallel. (5) CXT: In CT abdominal conditions, there is calcification at the tip of the xiphoid process. (6) HRA: At the point of contact with the xiphoid process, the rectus abdominis is thicker than the surrounding rectus abdominis. For cases with multiple tips of the xiphoid process, measurements for each sign were taken at the tip with a positive PXTRA. In cases where this criterion was not applicable, measurements were taken at the longest tip.

As additional items for CT signs, we recorded the imaging range of the CT, the slice thickness of the original CT images, and anatomical features of the xiphoid process (shape, number of tips, xiphoid foramen, fractures, length). Regarding the shape of the xiphoid process, forward curvature was denoted as "F", backward curvature as "B", and the curvature sequence from the base of the xiphoid process was represented (e.g., if the sequence is F→B→F, it is labeled as "FBF type"), defining it based on the number and direction of curvatures (Fig 4). All observations were assessed by the same evaluator.

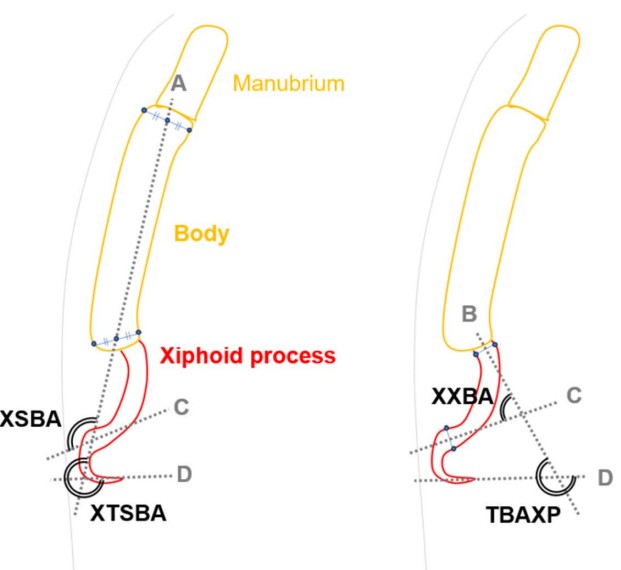

**Fig 2. The definition of the xiphisternal angle.** A: A line connecting the midpoint of the cranial end and the midpoint of the caudal end of the sternal body. B: A tangent line to the midline of the base of the xiphoid process (excluding the xiphisternal joint). C: A tangent line to the midline at the maximum ventral curvature of the xiphoid process. D: A tangent line at the tip of the xiphoid process. XSBA: The angle between line A and line C, with line A as the baseline. XXBA: The angle between line B and line C, with line B as the baseline. XTSBA: The angle between line A and line D, with line A as the baseline. TBAXP: The angle between line B and line D, with line B as the baseline.

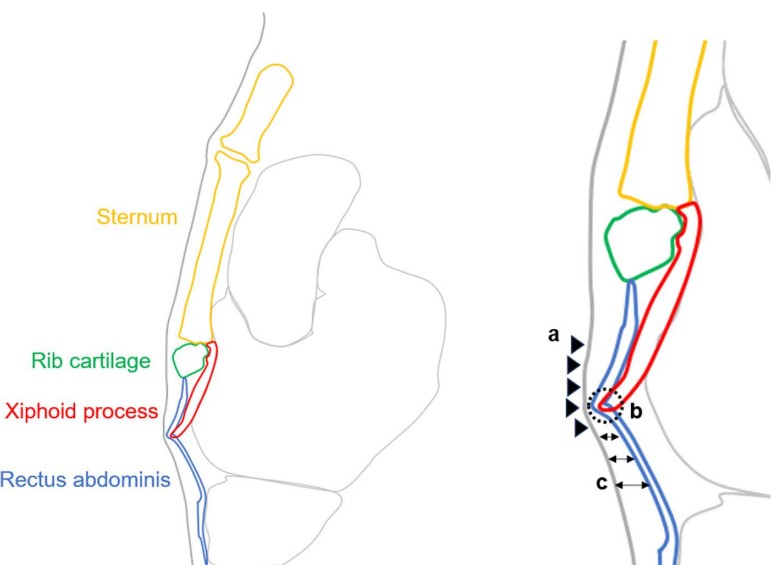

**Fig 3. Illustration of a CT sagittal image around the xiphoid process and descriptions of soft tissue compression signs and anatomical features of the xiphoid tip.** The xiphoid process compresses the rectus abdominis and skin ventrally, leading to positive signs for ASRA and ASSS (a). While the xiphoid tip protrudes into the rectus abdominis, resulting in a positive PXTRA sign, there's no rectus abdominis thickening, hence a negative HRA sign (b). Subcutaneous tissue thins due to xiphoid pressure, resulting in a positive TST sign (c).

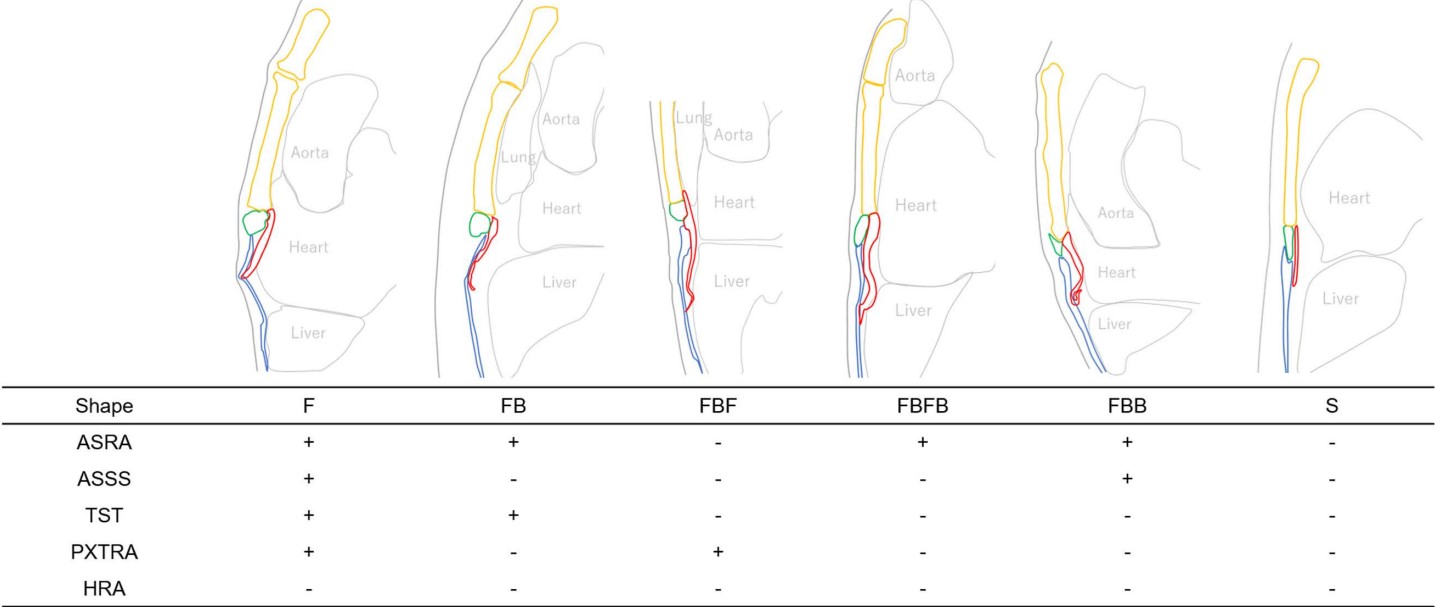

| Shape | F | FB | FBF | FBFB | FBB | S |
|-------|---|----|----|------|-----|---|
| ASRA | + | + | - | + | + | - |
| ASSS | + | - | - | - | + | - |
| TST | + | + | - | - | - | - |
| PXTRA | + | - | + | - | - | - |
| HRA | - | - | - | - | - | - |

**Fig 4. Examples of the xiphoid process shape, soft tissue compression signs, and anatomical findings of the xiphoid tip (exclude CXT).** The shape of the xiphoid process is defined by noting any bends from its base, using 'F' for forward bends and 'B' for backward bends. If it is straight, we denote it as 'S'.

### Statistical analyses

Each variable will be compared between case and control groups. For binary variables, odds ratios for case/control group will be calculated. For continuous variables, a two-sample t-test will be conducted. Data aggregation will be performed using Microsoft Excel®, and data analysis will be carried out using EZR on R-commander (version 1.41 for Windows) [11].

### Ethics approval

This study involves human participants and was approved by Kameyama Municipal Medical Center Medical Research and Ethics Committee (code No.2023092101). Informed consent was not obtained verbally; instead, it was obtained through written opt-out notices posted on the outpatient bulletin board at Kameyama Municipal Medical Center. Those who rejected were excluded. An anonymization table was created separately from the database, and all data was completely anonymized before analysis. No personally identifiable information, such as names or addresses, was entered into the database.

## Results

### Baseline characteristics

During the study period, 1560 patients presented with chest pain or abdominal pain. Among them, 11 were diagnosed with xiphodynia, with 10 meeting all criteria for xiphodynia. Nine of these, excluding one who did not undergo a CT scan including the xiphoid process, were assigned to the case group. The control group comprised 321 individuals who were diagnosed with other causes of pain and underwent CT scans including the xiphoid process.

Fig 5 presents the demographic characteristics of patients in both the case and control groups. There were no significant differences in age and gender between the case and control groups. In the control group, the most common causes of pain were gastrointestinal disorders such as intestinal obstruction (27 cases), ischemic colitis (19 cases), appendicitis (19 cases), diverticulitis (17 cases), totaling 103 cases, followed by renal and urinary tract disorders such as ureteral stones (54 cases), totaling 64 cases.

### Xiphisternal angle (Fig 6)

There were no significant differences observed between the two groups in any of the four defined angles. In the control group, 21 cases had the xiphoid process base not captured in the imaging, thus preventing measurement of all four angles. Additionally, XSBA and XSTBA measurements were not possible in 175 cases due to insufficient imaging of the sternum; these cases were excluded from the analysis for each respective angle. The range of xiphisternal angles in the case group was XSBA 104–174°, XXBA 91–163°, XTSBA 129–298°, and TBAXP 135–295°. In the control group, the ranges were XSBA 10–230°, XXBA 7–197°, XTSBA 10–379°, and TBAXP 7–363°.

### Compression signs of soft tissues (Fig 6)

There were no significant differences observed between the two groups in any of the ASRA, ASSS, or TST signs.

### Anatomical signs and morphology of the xiphoid process (Fig 6)

There were no significant differences observed between the two groups in any of the PXTRA, CXT, or HRA signs. Regarding the morphology of the xiphoid process, the FB type was the most common in both the case and control groups, followed by the F type, together comprising approximately 90% of the total. Within the control group, there was one case of complete absence of the xiphoid process and one case of xiphoid process fracture with bone fragment detachment. The tip of the xiphoid process was single in 269 cases, double in 59 cases, and triple in one case. Xiphoid foramen was observed in 49 cases. There were no significant differences observed between the case and control groups in the

| | Case(n=9) | Control(n=321) | P value |
|---|---|---|---|
| Background | | | |
| Age(year), mean(SD) | 71(15) | 68(21) | 0.63 |
| Male, n(%) | 4(44.4) | 150(46.7) | 1 |
| Location of pain, n(%) | | | |
| Chest pain | 3(33.3) | 43(13.4) | |
| Abdominal pain | 8(88.9) | 286(89.1) | |
| Cause of pain, n(%) | | | |
| Lung disease | - | 10(3.1) | |
| Heart disease | - | 17(5.3) | |
| Gastroduodenal disease | - | 20(6.2) | |
| Intestinal disease | - | 103(44.6) | |
| Liver disease | - | 5(1.6) | |
| Biliary tract disease | - | 33(10.3) | |
| Pancreas disease | - | 3(0.9) | |
| Uterine ovarian disease | - | 15(4.7) | |
| Renal urinary tract disease | - | 64(19.9) | |
| Cancer | - | 25(7.8) | |
| Musculoskeletal disease | - | 15(4.7) | |
| Other disease | - | 11(3.4) | |
| CT imaging range, n(%) | | | |
| Chest | 7(77.8) | 116(36.1) | |
| Abdomen | 4(44.4) | 302(94.1) | |
| CT slice thickness, n(%) | | | |
| 1.25mm | 9(100) | 168(52.3) | |
| 2.5mm | 0(0) | 14(4.4) | |
| 5mm | 0(0) | 139(43.3) | |

**Fig 5. Characteristics of case and control groups.**

ratios of these morphologies. The length of the xiphoid process showed no significant difference between the case group (60.6±11.1) and the control group (60.4±15.0).

### Relationship between patient demographics and anatomical features of the xiphoid process

We investigated the association between age, gender, and xiphisternal angle (XXBA), as well as between age, gender, and xiphoid process length. XXBA, which is less affected by tip inversion and has a larger sample size, was utilized. Outliers for each variable were excluded. We found a very weak correlation between age and XXBA (correlation coefficient -0.16, 95% CI: -0.27 to -0.05, p=0.005), but no significant correlation between age and xiphoid process length (correlation coefficient -0.04, 95% CI -0.16 to 0.09, p=0.56). The scatter plots for each variable are shown in Fig 7. Additionally, there was no significant difference between gender and XXBA (males: 138.6±25.8° vs. females: 141.4±26.1°, p=0.34),

| | Case(n=9) | Control(n=321) | OR(95%CI) | P value |
|---|---|---|---|---|
| **Xiphisternal angle, mean(SD) *** | | | | |
| XSBA | 142.1(22.3) | 144.7(29.2) | | 0.79 |
| XXBA | 144.3(17.5) | 139.9(26.1) | | 0.62 |
| XTSBA | 204.2(59.4) | 194.6(58.2) | | 0.63 |
| TBAXP | 206.4(58.3) | 185.2(48.9) | | 0.2 |
| **Soft tissue compression signs, n(%)** | | | | |
| ASRA | 7(77.8) | 211(65.7) | 1.82(0.34 to 18.26) | 0.72 |
| ASSS | 1(11.1) | 113(35.2) | 0.23(0.01 to 1.76) | 0.17 |
| TST | 7(77.8) | 200(62.3) | 2.12(0.39 too 21.17) | 0.49 |
| **Anatomical signs of xiphoid tip, n(%)** | | | | |
| PTXRA | 2(22.2) | 55(17.1) | 1.38(0.14 to 7.51) | 0.66 |
| CXT | 0(0) | 77(24.0) | 0(0 to 21.97) | 0.12 |
| HRA | 0(0) | 8(2.5) | 0(0 to 23.95) | 1 |
| **Anatomical characteristics \*\*** | | | | |
| Shape of xiphoid process, n(%) | | | | 0.61 |
| S | 0(0) | 12(3.7) | | |
| F | 3(33.3) | 72(22.4) | | |
| FB | 5(55.6) | 213(66.4) | | |
| FBF | 1(11.1) | 19(5.9) | | |
| FBB | 0(0) | 2(0.6) | | |
| FBFB | 0(0) | 2(0.6) | | |
| Number of tips, n(%) | | | | 0.69 |
| 1 | 7(77.8) | 262(81.6) | | |
| 2 | 2(22.2) | 57(17.8) | | |
| 3 | 0(0) | 1(0.3) | | |
| Xiphoid foramen, n(%) | 2(22.2) | 47(14.6) | | 0.63 |
| Length, mean(SD) | 60.6(11.1) | 60.4(15.0) | | 0.98 |

**Fig 6. Comparison of CT signs between case and control groups.** * We excluded cases where all four angles couldn't be measured due to the xiphoid process base not being within the imaging range in 21 controls. Additionally, 175 controls were excluded from XSBA and XSTBA measurements because the sternum wasn't adequately imaged. ** We omitted one case of xiphoid process absence from the description but included it in the denominator for probability calculations.

whereas a significant difference was observed between gender and xiphoid process length (males: 61.5±11.7 cm vs. females: 57.7±10.6 cm, p=0.005).

## Discussion

In our study, we investigated CT signs useful for diagnosing xiphodynia. However, no significant differences were observed between the two groups in terms of the xiphisternal angle, which has been considered useful in previous studies. Additionally, among the compression signs of soft tissues anterior to the xiphoid process and anatomical signs at the tip of the xiphoid process, no signs were found to have odds ratios indicative of diagnostic utility.

One reason why we did not find a significant difference in the xiphisternal angle might be that, unlike previous research, even the control group had smaller angles. In Maigne et al.'s study, the control group's angle averaged 172±15° [6], while

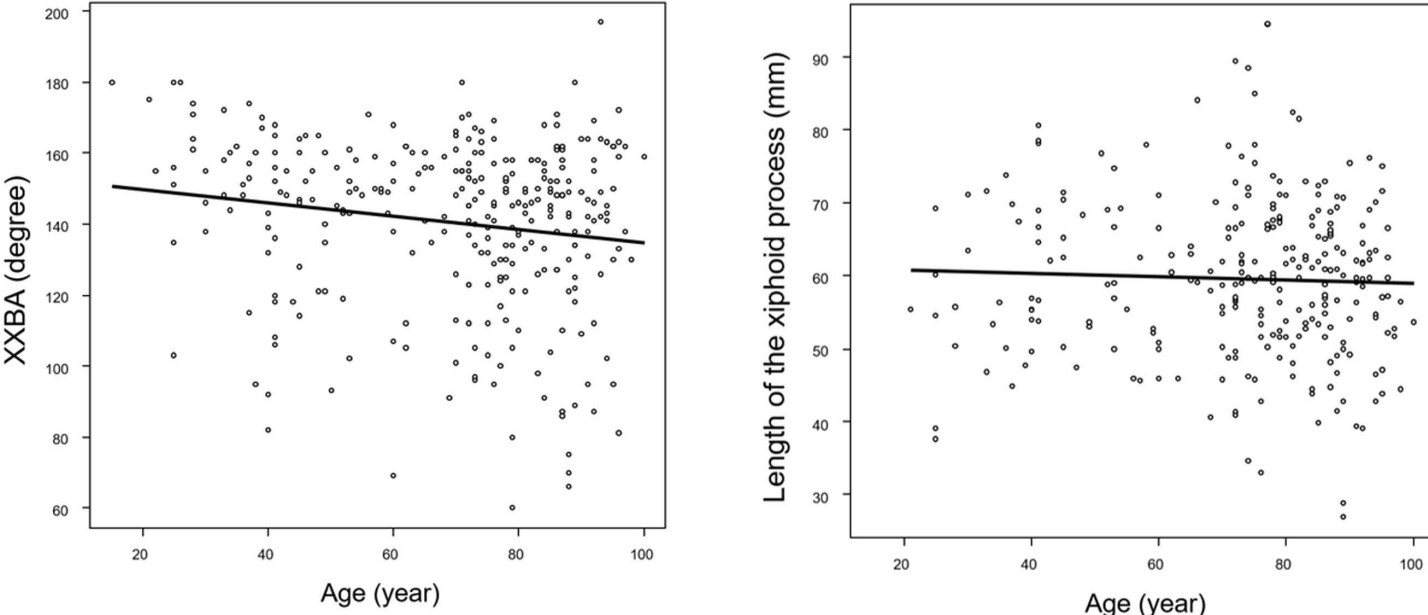

**Fig 7. A scatter plot depicting the correlation between age and the length of the xiphoid process, as well as the correlation between age and the xiphisternal angle.**

in our study, the XSBA and XXBA in the control group were around 140°. This could be because of differences in the age distribution of the control groups. In Maigne et al.'s study, the control group consisted of 60 individuals without xiphodynia, with an average age of 55±2 years [6]. On the other hand, the control group in our study had an average age of 68±21 years, indicating older age and greater variability compared to the previous study. The weak but positive correlation observed between XXBA and age in our study suggests that the older age of our control group may have contributed to lower xiphisternal angles. Secondly, it is well-established that skeletal differences exist among races, and it's possible that racial differences within the control group influenced the angulation. Although we did not find prior studies discussing racial differences in the xiphoid process, research has reported skeletal variations between Western and Eastern populations in cranial and pelvic bones [12]. Since our study was conducted in Japan and Maigne et al.'s study was conducted in France, there is a possibility that differences in skeletal morphology due to race could have influenced the angulation. Thirdly, the lack of clear definition for the xiphisternal angle in previous studies might have contributed to this outcome. While such differences exist between our study and previous research, our study suggests that a decrease in xiphisternal angle is common even in healthy individuals. Perhaps, our focus on xiphisternal angles has been limited to instances of xiphodynia, leading to a misconception that decreased xiphisternal angles are specific findings for xiphodynia.

Furthermore, in our study, the xiphisternal angle in the case group was larger than in previous studies. In prior research, many cases with decreased xiphisternal angle on CT images have been reported. Maigne et al. reported three cases with angles of 105°, 120°, and 135° [6], Patel et al. reported one case with an angle of 100° [13], Ono et al. reported one case with an angle of 128° [7], Ishizuka et al. reported one case with an angle of 133° [8], and Ismail et al. reported one case with an angle of 138.4° [14]. Additionally, although not explicitly stated regarding the angles, there are numerous reports of cases with images showing a prominently upwardly curved xiphoid process [9,15–17]. Conversely, in case series literature, there are more reports of cases with larger xiphisternal angles. Bakens et al. reported 19 cases who underwent xiphoidectomy for xiphodynia, with 15 cases having xiphisternal angle measurements ranging from 103° to 177°, with an average of 149.1±19.5° [18]. Dorn et al. reported 11 cases who underwent xiphoidectomy for xiphodynia, with all cases

having xiphisternal angle measurements ranging from 120° to 208°, with an average of 156.1±23.8° [19]. While there may be slight variations in angle definitions, the xiphisternal angles in our case group were in the early 140s, which is similar to those observed in the two case series. This similarity suggests that a significant reduction in the xiphisternal angle is not a common feature in many cases of xiphodynia. Therefore, the absence of a decrease in xiphisternal angle should not lead to underestimating the possibility of xiphoidynia, and xiphisternal angle may not be a useful diagnostic tool for xiphodynia.

Regarding compression signs of soft tissues, no signs indicating useful odds ratios for diagnosis were identified in our study. Xiphodynia is a relatively rare condition with low awareness among physicians, and studies conducted solely in single institutions have limited case numbers. In the future, collaborative multicenter studies with increased case numbers are desired.

One of the achievements of our study is the acquisition of new insights into the anatomical features of the xiphoid process and the CT interpretation methods. Initially, we hypothesized that inflammation-induced thickening of the xiphoid process was caused by its tip penetrating the rectus abdominis muscle, and thus, they incorporated hypertrophy of the rectus abdominis (HRA) into the study. However, it was found that the cartilaginous portion of the xiphoid process tip was in contact with the rectus abdominis muscle, and both structures had similar CT values, leading to the appearance of rectus abdominis hypertrophy. It was realized that a slice thickness of 5mm was insufficient for distinguishing this example, necessitating imaging with a slice thickness of 1.25mm. Additionally, since calcification at the tip of the xiphoid process (CXT positive) was observed in only 23% of cases, there is a possibility of misinterpreting the calcified portion as the true tip of the xiphoid process. To confirm the non-calcified portion, conditions in the abdominal sagittal plane were found to be suitable. When assessing the anatomical features of the xiphoid process tip on CT, imaging with abdominal conditions of 1.25mm slice thickness or less and confirmation using sagittal reconstruction images are necessary.

Additionally, it was found that the anatomical feature of the xiphoid process known as the "FB type," where the xiphoid process curves forward and then reverses backward, accounted for more than 60% of cases. In a study by Akin et al., which assessed the shape of the xiphoid process using MDCT images of 500 individuals, a shape referred to as the "S-shape," where the xiphoid process curved forwardly and then reversed backwardly, was identified in 4 cases (0.8%) [20]. Although the study did not discuss events where the xiphoid process reversed, multiple figures presented as cases of forwardly curved xiphoid process in the same literature confirm reversing after contact between the cartilage part of the xiphoid process tip and the rectus abdominis muscle. Moreover, numerous studies on xiphodynia have presented images showing reversal [13,18,21], indicating that reversal of the xiphoid process tip is a common occurrence regardless of the presence of xiphodynia. In our study, no cases exhibited a shape where the xiphoid process first curved backwardly (toward the heart) from the base (B or BF type). There were also few cases with shapes like FBF or FBFB where the xiphoid process reverses forwardly again, and in such cases, the liver was positioned backwardly to the xiphoid process (Fig 4). From these observations, we believe that the xiphoid process bends forwardly due to pressure from internal organs such as the heart or liver, and backwardly reverses due to pressure from abdominal wall like the rectus abdominis. We speculate that the positive correlation observed between age and xiphisternal angle in our study may be attributed to worsening kyphosis with age, leading to increased pressure from internal organs on the xiphoid process. It is noteworthy that there was no difference in the shape of the xiphoid process between the case and control groups, suggesting that the likelihood of curvature or reversal of the xiphoid process being associated with xiphodynia is low. However, considering the common occurrence of reversal of the xiphoid process tip, it is inappropriate to use TBAXP or XTSBA, which use the xiphoid process tip as the baseline for xiphisternal angle measurements. Instead, it is advisable to use XXBA or XSBA, which use the baseline where the xiphoid process is most forwardly curved or just before it contacts the rectus abdominis muscle.

## Limitation

Firstly, since this study is a case-control study within a cohort, the control group does not consist of individuals who are non-xiphodynia. Diagnosis of xiphodynia is typically based on medical history, physical examination, and exclusion

diagnosis, which are minimally invasive, and the control group should comprise individuals without the condition ideally. Additionally, although there is a report indicating a prevalence of xiphodynia in hospitalized cases at 2% [22], the exact prevalence is unknown, and there is a possibility that a considerable number of xiphodynia patients may be included in the control group. In the future, investigations into the prevalence are necessary, and if a study is to be conducted with an increased sample size, using a prospective control group without the disease would be appropriate. Secondly, the sample size was insufficient to compare findings such as compression signs of soft tissues and anatomical features of the xiphoid process tip. Xiphodynia is a relatively rare condition, and evaluation in a single institution has its limitations, thus collaborative research across multiple institutions to increase the sample size is desirable. Thirdly, there is a possibility that the definition of the sternal baseline used for measuring the xiphisternal angle was not appropriate. While a linear sternum poses no issues, if the sternum is arc-shaped, the angle may vary slightly depending on whether the baseline is defined as in this study or tangentially to the circular sternum. Since tangential direction assessment may lead to variability among evaluators, we opted for our original definitions for this study, which is easier to judge as a landmark. Fourthly, regarding compression signs of soft tissues, there is a potential for variation depending on the position, but CT examination is limited to the supine position, possibly missing patients with compression signs appearing in the upright or seated positions. Ultrasound examinations can be performed regardless of position, suggesting their potential usefulness in evaluating compression signs of soft tissues, which warrants further investigation.

## Conclusion

The xiphisternal angle was suggested to be ineffective in the imaging diagnosis of xiphodynia. The compression signs of soft tissues and anatomical features of the xiphoid process tip set in this study warrant a reassessment with an increased number of cases. Additionally, xiphoid process reversing is frequently observed regardless of the presence of xiphodynia, with forward curvature likely influenced by pressure from internal organs and backward curvature by pressure from the abdominal wall. For confirming these, CT conditions involving axial slices of 1.25mm thickness or less in the abdominal region are recommended.

## Author contributions

**Conceptualization:** Ryosuke Ono.

**Formal analysis:** Ryosuke Ono.

**Methodology:** Ryosuke Ono.

**Writing – original draft:** Ryosuke Ono.

**Writing – review & editing:** Ken Horibata.

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
