## [Decision Letter · Decision Letter 0]

1 Jul 2024

PONE-D-24-16507A case-control study evaluating CT signs of xiphoid process associated with xiphodyniaPLOS ONE

Dear Dr. Ono,

Thank you for submitting your manuscript to PLOS ONE. After careful consideration, we feel that it has merit but does not fully meet PLOS ONE’s publication criteria as it currently stands. Therefore, we invite you to submit a revised version of the manuscript that addresses the points raised during the review process.

Since the Reviewers carefully reviewed the article, your manuscript needs further revision. 

We look forward to receiving your revised manuscript.

Kind regards,

Roham Borazjani

Academic Editor

PLOS ONE

Journal Requirements:

3. In the online submission form, you indicated that your data will be submitted to a repository upon acceptance.  We strongly recommend all authors deposit their data before acceptance, as the process can be lengthy and hold up publication timelines. Please note that, though access restrictions are acceptable now, your entire minimal  dataset will need to be made freely accessible if your manuscript is accepted for publication. This policy applies to all data except where public deposition would breach compliance with the protocol approved by your research ethics board. If you are unable to adhere to our open data policy, please kindly revise your statement to explain your reasoning and we will seek the editor's input on an exemption.

Reviewers' comments:

Reviewer's Responses to Questions

**Comments to the Author**

1. Is the manuscript technically sound, and do the data support the conclusions?

Reviewer #1: Yes

Reviewer #2: Yes

2. Has the statistical analysis been performed appropriately and rigorously? 

Reviewer #1: Yes

Reviewer #2: Yes

3. Have the authors made all data underlying the findings in their manuscript fully available?

Reviewer #1: Yes

Reviewer #2: Yes

4. Is the manuscript presented in an intelligible fashion and written in standard English?

Reviewer #1: Yes

Reviewer #2: Yes

5. Review Comments to the Author

Reviewer #1: This is an interesting study of the xiphoid process, has been evaluated by CT scan, in the diagnosing xiphodynia. concluded that The xiphoid process sternal angle is not useful for diagnosing xiphodynia. The paper is well written and ethical issues are considered appropriately.

Reviewer #2: First, I would like to congratulate the authors on an interesting study and the efforts put forth to answer a rare clinical conundrum. Overall, the manuscript is well written, easy to read, and understand. There are a few minor syntax errors that could be corrected. In addition, would consider using the term study group rather than case group in the abstract as it reads better. In some instance, you may want to consider using the term xiphosternal angulation as oppose to angle as this represents a phenomenon rather than an anatomic finding. In lines 169, 171, and 173, recommend specification of the origin of the line, for example, "from the anterior abdominal wall to..." as any line has to have two points of reference. In the table on page 10, please reduce the size of the table 2 to conform to the page as it runs over the margin. In line 359 and 360 on page 13, the phrase is not clear, "although not explicitly stated the angles?" Please revise, perhaps, "although not explicitly stated regarding the angles..." Again, enjoyed reading and reviewing your submission.

6. PLOS authors have the option to publish the peer review history of their article (what does this mean?). If published, this will include your full peer review and any attached files.

Reviewer #1: **Yes: **Gholam Reza Raissi MD

Reviewer #2: **Yes: **Omar K Danner, MD, MBA, FACS, FCCM

---

## [Author Response · Author response to Decision Letter 1]

9 Jul 2024

Dear Reviewers,

I would like to express my sincere gratitude for the thorough review and constructive feedback you provided on my manuscript submitted to PLOS ONE. Your insightful comments have been invaluable in improving the quality and clarity of the manuscript.

I have carefully considered each of your suggestions and incorporated them into the revised version. Your expertise and thoughtful input have significantly contributed to the overall enhancement of the paper.

Thank you once again for your time, effort, and valuable contributions to the advancement of this work. Your dedication to the peer review process is greatly appreciated.

Best regards,

Corresponding Author: Ryosuke Ono

Department of Community Medicine, Kameyama

rikusan2005@yahoo.co.jp

Journal Requirements:

1. When submitting your revision, we need you to address these additional requirements. Please ensure that your manuscript meets PLOS ONE's style requirements, including those for file naming. The PLOS ONE style templates can be found at

→We have confirmed that the submitted file meets PLOS ONE's style requirements.

2. Thank you for uploading your study's underlying data set. Unfortunately, the repository you have noted in your Data Availability statement does not qualify as an acceptable data repository according to PLOS's standards. At this time, please upload the minimal data set necessary to replicate your study's findings to a stable, public repository (such as figshare or Dryad) and provide us with the relevant URLs, DOIs, or accession numbers that may be used to access these data. For a list of recommended repositories and additional information on PLOS standards for data deposition, please see https://journals.plos.org/plosone/s/recommended-repositories.

3. In the online submission form, you indicated that your data will be submitted to a repository upon acceptance. We strongly recommend all authors deposit their data before acceptance, as the process can be lengthy and hold up publication timelines. Please note that, though access restrictions are acceptable now, your entire minimal dataset will need to be made freely accessible if your manuscript is accepted for publication. This policy applies to all data except where public deposition would breach compliance with the protocol approved by your research ethics board. If you are unable to adhere to our open data policy, please kindly revise your statement to explain your reasoning and we will seek the editor's input on an exemption.

→We have uploaded the data to Dryad. Below are the Reviewer link and DOI for your reference.

ReviewerURL: https://datadryad.org/stash/share/12RMAxde1Qt6QDs9SCraPdbrGHPpKrE1-DYXcfPtJxU

doi:10.5061/dryad.ghx3ffbxw

→We have confirmed that all references are publicly accessible. References 2 and 15 are not indexed in PubMed.

Reviewer #1: This is an interesting study of the xiphoid process, has been evaluated by CT scan, in the diagnosing xiphodynia. concluded that The xiphoid process sternal angle is not useful for diagnosing xiphodynia. The paper is well written and ethical issues are considered appropriately.

→Thank you for your evaluation. It has become an encouragement for my future research career.

Reviewer #2: First, I would like to congratulate the authors on an interesting study and the efforts put forth to answer a rare clinical conundrum. Overall, the manuscript is well written, easy to read, and understand. There are a few minor syntax errors that could be corrected. In addition, would consider using the term study group rather than case group in the abstract as it reads better.

→Thank you very much for your feedback. We have made some corrections based on it.

In some instance, you may want to consider using the term xiphosternal angulation as oppose to angle as this represents a phenomenon rather than an anatomic finding.

→In the discussion, we changed "angle" to "angulation" in two instances where it conveyed the meaning of "a phenomenon rather than an anatomic finding." Upon further investigation into the meanings of "angulation" and "angle," as you pointed out, it seems that "xiphisternal angulation" would convey a more appropriate nuance than "xiphisternal angle." However, since "xiphisternal angle" has been consistently used in previous literatures, we have opted to adopt this term in our study as well.

In lines 169, 171, and 173, recommend specification of the origin of the line, for example, "from the anterior abdominal wall to..." as any line has to have two points of reference.

→I am debating whether to make the changes as you suggested. For the B-line, I considered modifying it from "A tangent line to the midline of the base of the xiphoid process" to "A line perpendicular to the midpoint between the ventral and dorsal sides at the base of the xiphoid process." However, I believe this expression would not be suitable for the D-line because the thickness at the tip of the xiphoid process is approximately zero. Moreover, considering that tangency is defined as "a straight line passing through two points infinitely close on a curve," it seems unnecessary to strictly define two points.

Ultimately, the key is how to define these lines to enhance the reproducibility of the study. Since we initially adopted tangents as curves rather than midlines of thickness, we prefer to use the original description. If changes are still preferable based on the considerations above, I am willing to make adjustments. Your feedback on this matter would be appreciated.

In the table on page 10, please reduce the size of the table 2 to conform to the page as it runs over the margin.

→We have made adjustments to fit within the framework.

In line 359 and 360 on page 13, the phrase is not clear, "although not explicitly stated the angles?" Please revise, perhaps, "although not explicitly stated regarding the angles..."

→As you suggested, we have made the changes.

---

## [Decision Letter · Decision Letter 1]

12 Aug 2024

PONE-D-24-16507R1A case-control study evaluating CT signs of xiphoid process associated with xiphodyniaPLOS ONE

Dear Dr. Ono,

Thank you for submitting your manuscript to PLOS ONE. After careful consideration, we feel that it has merit but does not fully meet PLOS ONE’s publication criteria as it currently stands. Therefore, we invite you to submit a revised version of the manuscript that addresses the points raised during the review process.

We look forward to receiving your revised manuscript.

Kind regards,

Roham Borazjani

Academic Editor

PLOS ONE

Journal Requirements:

Additional Editor Comments:

Thanks for addressing all the reviewer comments. Please make sure that your manuscript structure follows the PLOS ONE formatting sample.

Please submit an unstructured abstract.

revise your references, figures, and tables to follow the PLOS ONE formatting sample (find the attached PDF)

Reviewers' comments:

Reviewer's Responses to Questions

**Comments to the Author**

1. If the authors have adequately addressed your comments raised in a previous round of review and you feel that this manuscript is now acceptable for publication, you may indicate that here to bypass the “Comments to the Author” section, enter your conflict of interest statement in the “Confidential to Editor” section, and submit your "Accept" recommendation.

Reviewer #1: All comments have been addressed

Reviewer #2: All comments have been addressed

2. Is the manuscript technically sound, and do the data support the conclusions?

Reviewer #1: Yes

Reviewer #2: Yes

3. Has the statistical analysis been performed appropriately and rigorously? 

Reviewer #1: Yes

Reviewer #2: Yes

4. Have the authors made all data underlying the findings in their manuscript fully available?

Reviewer #1: Yes

Reviewer #2: Yes

5. Is the manuscript presented in an intelligible fashion and written in standard English?

Reviewer #1: Yes

Reviewer #2: Yes

6. Review Comments to the Author

Reviewer #1: The paper is a good work regarding xphodnia , I had no specific comment in review and can be accepted for publication.

Reviewer #2: Thank you making the updates and revisions and your efforts. This is a very well written manuscript. It sets the foundation for a multi-institutional study to further delineate the subject matter. Recommend changing the last reference to the appropriate case as the authors names do not need to be in all capitalized letter. Otherwise, no additional changes were identified or are recommended.

7. PLOS authors have the option to publish the peer review history of their article (what does this mean?). If published, this will include your full peer review and any attached files.

Reviewer #1: **Yes: **Golam Reza Raissi

Reviewer #2: No

---

## [Author Response · Author response to Decision Letter 2]

25 Aug 2024

Dear Editors and Reviewers,

I am writing to submit the revised version of my manuscript titled “[Title of Your Manuscript]” for reconsideration by PLOS ONE. I sincerely appreciate the additional feedback and comments provided by the reviewers and editorial team following my previous submission. Their continued guidance has been instrumental in refining the manuscript further.

In this revision, I have made every effort to ensure that the manuscript adheres to the submission guidelines of PLOS ONE. Notably, the tables have been converted to figures in accordance with the journal’s requirements. I hope that my submission is now fully compliant with the guidelines.

Thank you once again for your time, effort, and valuable input in reviewing this work. I look forward to your feedback on the revised manuscript.

Best regards,

Corresponding Author: Ryosuke Ono

Department of Community Medicine, Kameyama

rikusan2005@yahoo.co.jp

Journal Requirements:

→We have corrected the author names in reference 23, which were previously in uppercase.

Additional Editor Comments:

Thanks for addressing all the reviewer comments. Please make sure that your manuscript structure follows the PLOS ONE formatting sample.

Please submit an unstructured abstract.

revise your references, figures, and tables to follow the PLOS ONE formatting sample (find the attached PDF)

→We have revised the abstract to make it less structured. Additionally, we have converted all tables to figures. The font in the figures has been changed to the specified one, and PACE has also been used.

Reviewer #2: Recommend changing the last reference to the appropriate case as the authors names do not need to be in all capitalized letter.

→We hadn't noticed this until it was pointed out, so we appreciate your help. We have made the necessary corrections.

---

## [Decision Letter · Decision Letter 2]

25 Apr 2025

A case-control study evaluating CT signs of xiphoid process associated with xiphodynia

PONE-D-24-16507R2

Dear Dr. Ono,

We’re pleased to inform you that your manuscript has been judged scientifically suitable for publication and will be formally accepted for publication once it meets all outstanding technical requirements.

Kind regards,

Richa Gupta

Academic Editor

PLOS ONE

Additional Editor Comments (optional):

In this study, the authors have undertaken a thorough and comprehensive examination of the diagnostic criteria for xiphodynia. They have invested significant effort into addressing all the necessary revisions and improvements suggested by the reviewers and the editorial committee. Each amendment has been carefully and precisely implemented, ensuring that the article meets the standards of academic rigor and clarity. Given the depth of their work and the meticulous revisions made in response to the feedback, I wholeheartedly recommend the ACCEPTANCE of this article for publication."

Please find attached reviewer’s comments:

REVIEWER 1 – ACCEPT

This is an interesting study of the xiphoid process, has been evaluated by CT scan, in the diagnosing xiphodynia. concluded that The xiphoid process sternal angle is not useful for diagnosing xiphodynia. The paper is well written and ethical issues are considered appropriately.

Reviewers' comments:

Reviewer's Responses to Questions

**Comments to the Author**

1. If the authors have adequately addressed your comments raised in a previous round of review and you feel that this manuscript is now acceptable for publication, you may indicate that here to bypass the “Comments to the Author” section, enter your conflict of interest statement in the “Confidential to Editor” section, and submit your "Accept" recommendation.

Reviewer #1: All comments have been addressed

2. Is the manuscript technically sound, and do the data support the conclusions?

Reviewer #1: Yes

3. Has the statistical analysis been performed appropriately and rigorously? 

Reviewer #1: Yes

4. Have the authors made all data underlying the findings in their manuscript fully available?

Reviewer #1: Yes

5. Is the manuscript presented in an intelligible fashion and written in standard English?

Reviewer #1: Yes

6. Review Comments to the Author

Reviewer #1: The manuscriptis is compliant with the guidelines and is now acceptable for publication. the language is clear and coorect.

7. PLOS authors have the option to publish the peer review history of their article (what does this mean?). If published, this will include your full peer review and any attached files.

Reviewer #1: **Yes: **GholamrReza Raissi

---

## [Editor Report · Acceptance letter]

PONE-D-24-16507R2

PLOS ONE

Dear Dr. Ono,

I'm pleased to inform you that your manuscript has been deemed suitable for publication in PLOS ONE. Congratulations! Your manuscript is now being handed over to our production team.

Kind regards,

on behalf of

Dr. Richa Gupta

Academic Editor

PLOS ONE